# Choosing the Best Sensor Fusion Method: A Machine-Learning Approach

**DOI:** 10.3390/s20082350

**Published:** 2020-04-20

**Authors:** Ramon F. Brena, Antonio A. Aguileta, Luis A. Trejo, Erik Molino-Minero-Re, Oscar Mayora

**Affiliations:** 1Tecnologico de Monterrey, Av. Eugenio Garza Sada 2501 Sur, Monterrey 64849, Mexico; ramon.brena@tec.mx; 2Facultad de Matemáticas, Universidad Autónoma de Yucatán, Anillo Periférico Norte, Tablaje Cat. 13615, Colonia Chuburná Hidalgo Inn, Mérida 97110, Mexico; 3Tecnologico de Monterrey, School of Engineering and Sciences, Carretera al Lago de Guadalupe Km. 3.5, Atizapán de Zaragoza 52926, Mexico; ltrejo@tec.mx; 4Instituto de Investigaciones en Matemáticas Aplicadas y en Sistemas—Sede Mérida, Unidad Académica de Ciencias y Tecnología de la UNAM en Yucatán, Universidad Nacional Autónoma de México, Sierra Papacal 97302, Mexico; erik.molino@iimas.unam.mx; 5Fandazione Bruno Kessler Foundation, 38123 Trento, Italy; omayora@fbk.eu

**Keywords:** optimal, data fusion, meta-data, sensor fusion

## Abstract

Multi-sensor fusion refers to methods used for combining information coming from several sensors (in some cases, different ones) with the aim to make one sensor compensate for the weaknesses of others or to improve the overall accuracy or the reliability of a decision-making process. Indeed, this area has made progress, and the combined use of several sensors has been so successful that many authors proposed variants of fusion methods, to the point that it is now hard to tell which of them is the best for a given set of sensors and a given application context. To address the issue of choosing an adequate fusion method, we recently proposed a machine-learning data-driven approach able to predict the best merging strategy. This approach uses a meta-data set with the Statistical signatures extracted from data sets of a particular domain, from which we train a prediction model. However, the mentioned work is restricted to the recognition of human activities. In this paper, we propose to extend our previous work to other very different contexts, such as gas detection and grammatical face expression identification, in order to test its generality. The extensions of the method are presented in this paper. Our experimental results show that our extended model predicts the best fusion method well for a given data set, making us able to claim a broad generality for our sensor fusion method.

## 1. Introduction

As fixed and wearable sensors become increasingly pervasive in settings, such as healthcare, where reliability and accuracy are critical, several sensors are frequently used in combination to increase the overall performance. We use the term multi-sensor data fusion to refer to a collection of methods used to deal with sensor relative weaknesses, such as sensor malfunction, imprecision, limited spatial coverage, and uncertainty [1]. The combined use of several sensors stems from the observation that, in many cases, one of the sensors strengths can compensate for the weaknesses of the others. Overall, sensor combination aims to reach better performance than a single sensor [2] because it can improve the signal-to-noise ratio, decrease uncertainty and ambiguity, and increase reliability, robustness, resolution, accuracy, and other desirable properties [3].

We restricted our attention to scenarios where the sensors are digital or their signal is converted to digital data, so that the raw numeric stream can be further processed by algorithms, like feature extraction, that have found widespread use as a result of the popularity and availability of data-analysis platforms, in languages like R [4], Python [5], etc. In data-driven methods, the features extracted from raw data coming from sensors are fed to the decision-making algorithms, such as classifiers [6]. Even in the restricted context of digital information integration for decision processes, many fusion methods have been developed, such as Multi-view stacking [7], AdaBoost [8], and Voting [9], to mention a few.

One problem that arises given such a variety of methods to choose from is that it is often not clear which approach is the best choice for the particular set of sensors and a given data stream [10]. Indeed, it is common to choose, by way of trial and error, among some of these methods or, worse, to settle for the best-known methods [11].

To address this problem, in a previous work we proposed an approach, called “Prediction of Optimal Fusion Method” (POFM), based on machine learning techniques, that predicts the optimal fusion method for a given set of sensors and data [11]. However, although the POFM method has demonstrated very good performance, it only covers eight fusion configurations (based on methods: Voting, Multi-view stacking, and AdaBoost), two types of sensors (accelerometers and gyroscopes) and a particular domain (recognition of simple human activities (SHA) [12]).

As described in our previous work, the POFM method principle is to first construct a meta-data data set in which each individual row (that we call “Statistical signature”) contains statistical features of one entire data set; so, the meta-data data set is a data set of features of data sets. For each row, we experimentally test which of the considered fusion methods is the best one, and this method becomes the label (“ground truth”) of that row. Then, we train a classifier with standard machine-learning methods. The prediction of the classifier should be the best fusion method, so that when it presents a data set not used in the training process, it comes up with a prediction of best fusion method: if the prediction is equal to the label, then it succeeds; otherwise, it fails. Standard performance measures can easily be calculated from the success/failure rates, such as accuracy, precision, etc. Clearly, as each data set becomes a row in the Statistical signature data set, we need to gather a large number of data sets in order to be able to train a classifier with machine-learning techniques.

This work aims to extend our POFM method, to predict the best fusion method in other domains, in addition to the recognition of SHA, such as gas prediction [13] and the recognition of grammatical facial expressions (GFEs) [14]; we aimed for additional domains as different as possible from SHA. In addition, this work incorporates other sensors in addition to accelerometers and gyroscopes (for example, depth cameras and gas sensors), and, consequently, a deeper exploration of the method was required. Thus, the value of this work is to validate our generality claim about the pertinence of the POFM method across different domains.

The structure of this document is as follows: after this introduction, in Section 2, we present the state-of-the-art. In Section 3, we show the proposed method; then, in Section 4, we present the experiments and the discussion of their results. Finally, in Section 5, we draw some conclusions and propose possible future work.

## 2. State of the Art

Multi-sensor fusion was initially used in the United States Navy during the 1970s as a method to tackle some military problems, such as to improve the accuracy of the Soviet Navy’s motion detection [15]. Currently, this method is used in various applications, such as the verification of machinery, diagnostics in medicine, robotics, image and video processing, and smart buildings [16].

We refer to multi-sensor fusion method as a way of using, together, the information coming from several sensors. This could be done by joining the features extracted from the data or by combining the decisions made from these features by several classifiers [17]. The goal of such a fusion is to get improved accuracy, signal-to-noise ratio, robustness and reliability, accuracy or other good properties than using a single sensor [2] or to reduce some undesirable issues, such as uncertainty and ambiguity [3].

With respect to the abstraction level of data processing, Multi-sensor fusion has been classified into three categories [6,18]: fusion at the data-level, fusion at the feature-level, and fusion at the decision-level. These categories are explained in the following:

Fusion at the data-level: Data coming from several sensors is simply aggregated or put together in a database or stream, resulting in a bigger information quantity. The implicit assumption is that merging several similar input data sources will result in more precise, informative, and synthetic fused data than the isolated sources [19]. The works centered around data-level fusion are mainly intended to reduce noise or to increase robustness [20,21,22].

Fusion at the feature-level: Multiple features are derived from the raw data sources, either from independent sensor nodes or by a single one with several sensors, and they can be combined into a high-dimensional feature vector, which is then used as input for pattern-recognition tasks by a classifier [23]. Different ways of combining the features have been proposed, depending on the type of application, so that these vectors with multidimensional characteristics can then be morphed into transformed vectors of joint characteristics, from which classification is then carried out [24]. Examples of feature-level fusion are: Feature Aggregation [14,25,26,27,28,29,30,31], Temporal Fusion [32], Support Vector Machine (SVM)-based multisensor fusion algorithm [33], and Data Fusion Location algorithm [18].

Fusion at the decision-level: This consists of combining the local results of several decision processes or classifiers into one global decision, which is normally a target class or tag [34]. Adaboost [8], Voting [9], Multi-view Staking [7], and Hierarchical Weighted Classifier [35] are examples of methods that use the combination of local decisions into a global one; this is also the case of the Ensemble of Classifiers [13,36], the Genetic Algorithm-based Classifier Ensemble Optimization Method [37] and the Hidden Markov Models (HMM)-SVM framework for recognition [38].

## 3. Method

In Figure 1, we present our approach to predict the best fusion strategy for a given data set that belongs to one of the domains considered (for example, the recognition of SHA [12], gas detection [13], or recognition of GFEs [14]). Our approach extends the POFM [11] method by adding the Generalization step to the Statistical signature data set stage (see Figure 1); the rest of the stages of the POFM method remain unchanged. Next, we describe the phases of the extended POFM (EPOFM) model.

### 3.1. Finding the Best Fusion Method

As can be seen in Figure 1, this stage aims, for a given data set, to statistically find the best sensor data fusion configuration of a group of eight [11]. These eight configurations, based on at least one of these three prediction methods: Random Forest Classifier (RFC) [39], Logistic Regression (LR) [40], and Decision Tree (CART) [41], are Feature Aggregation [25,26], Voting [9] (with three configurations), Multiple view stacking [7] (with three settings), and AdaBoost [8]. Next, we summarize the eight fusion configurations:

Aggregation of features. This configuration, which is classified within the type of fusion at the characteristic level, includes two steps: (1) combine by column the characteristics extracted from a given data set; and (2) train a prediction model (such as RFC) with these characteristics to learn to predict the classes of the mentioned data set.

Vote with shuffled features. This configuration, categorized as a decision-level merger, consists of three steps. (1) take the characteristics extracted from a given data set, combine them by columns, shuffle them, and divide them into three parts, each part for each instance of a machine learning algorithm (such as RFC). (2) Use three instances of this algorithm as estimators and Voting as an assembly method. (3) train Voting method with the features of step 1) to learn to predict the classes recorded in the given data set.

Vote. This configuration is the same as the previous configuration (Vote with shuffled features), with the exception that, in step 1), the features are not shuffled before dividing them into three parts, for later assignment to three instances of the machine learning algorithm (for example, RFC). Therefore, the remaining steps remain unchanged.

Voting with RCF, CART, and LR for all features. This configuration takes advantage of two types of fusion (feature level and decision level) as follows: (1) extract the characteristics of a given data set and combine them by columns (feature level merge); (2) define RFC, CART, and LR as estimators and the voting algorithm as an ensemble method (for a decision-level fusion); and (3) train Voting method, giving each estimator all the characteristics of step 1), to learn to predict the classes recorded in the data set provided.

Multi-View Stacking with shuffled features. This configuration is based on the fusion at the decision level. It consists of the following steps: (1) obtain the features extracted from a given data set, combined them by columns, shuffled them, and divided them into three parts; (2) define three instances of a prediction model (for example, RFC) as the base classifiers; (3) train each instance of the base classifier with some of these three parts of features, and combine by column the predictions of these instances; and (4) define a machine learning algorithm (such as RFC) as meta-classifier (for a decision-level fusion), to train it with the combined prediction of step 3), with the goal that it learns to predict the labels of the provided data set.

Multi-View Stacking. This configuration is very similar to the previous setup (Multi-View Stacking with shuffled features); the only difference is that, in step 1), the characteristics are not shuffled before dividing them into three parts, for later assignment to three instances of the machine learning algorithm (for example, RFC). Therefore, the remaining steps remain unchanged.

Multi-View Stacking with RCF, CART and LR for all features. This configuration, that takes advantage of two fusion types (features level and decision level), includes the following steps: (1) get the features extracted from a given data set and combine them by columns (feature-level fusion); (2) define three classifiers (RFC, CART, or LR) as the base classifiers; (3) train each base classifier with these combined features and combine by column the predictions of these classifiers; and (4) define a prediction model (such as RFC) as meta-classifier (for a decision-level fusion), with the goal of train it with combined features of step 3), to learn to predict the classes recorded in provided data set.

AdaBoost. This configuration that exploits the advantages of two types of fusion (feature level and decision level) includes the following four steps: (1) take the characteristics extracted from the provided data set and combine them by columns (merging of the level of features); (2) define a classifier (for example, RFC) as an estimator; (3) select the Adaboost algorithm as an ensemble method (for a decision-level merger); and (4) train AdaBoost with the characteristics obtained in step 1) to learn how to infer the labels registered in the data set provided.

From these configurations and to achieve the objective of this stage, Friedman’s rank test [42] and Holm’s test [43] are conducted to see significant differences, in terms of accuracy, between Aggregation (defined as the baseline for comparison proposals) and each other fusion settings [11]. If the accuracy of more than one fusion configuration exhibits a significant difference concerning the accuracy of the Aggregation, the fusion configuration with the highest accuracy is taken as the best fusion strategy [11]. If none of the accuracies of the fusion configurations present a significant difference concerning the accuracy of the Aggregation, the Aggregation method is considered the best fusion strategy [11]. We use the Friedman range test (non-parametric test) with the corresponding posthoc tests (Holm test) because the Friedman test is safer and more robust than the parametric tests (such as ANOVA [44]), in the context of having to compare two or more prediction models in different data sets [45]. Although we do not directly compare classifiers, we do so indirectly, since merge methods are classifier-based. Therefore, it seems convenient to use the Friedman test. We use Holm’s posthoc test instead of another more complex posthoc test (for example, Hommel’s test [46]) because both types of tests show practically no difference in their results [45]. In addition, according to Holland [47], the Hommel test does not present a significant advantage over the Holm test.

### 3.2. Statistical Signature Dataset

The objective of this stage is to build a meta-data set with a labeled Statistical signature extracted from each data set [11] that belongs to one of the domains considered here. The idea behind this Statistical signature is that it can be used as a feature vector to train a prediction model (such as RFC) to learn how to predict the tag (the best fusion configuration) of a Statistical signature of a given data set [11]. Next, we present the procedure to create this meta-data set (see Figure 2), which since it integrates the Statistical signature of the data sets of different domains, we say that it is a generalized meta-data set.

As can be seen in Figure 2, in the Statistical signature extraction step, we first construct a Statistical signature of the features of each data set from each domain considered. The procedure to build these signatures consists of obtaining the mean (mean()), standard deviation (std()), the maximum value (max()), minimum value (min()), and 25th ($P_{25th}\left(\right)$), 50th ($P_{50th}\left(\right)$), and 75th ($P_{75th}\left(\right)$) percentiles, for each of the characteristics of those data sets [11]. For example, let *A* be a matrix representing one of the mentioned data sets, with dimensions *S* (rows, containing samples) and *F* (columns, with features), and let $a_{ij}$ be an element of *A*, where i=1,2,…,S and j=1,2,…,F. Then, *A* can be seen as the set of vectors $AV=\{f_{1},f_{2},\ldots ,f_{F}\}$, where $f_{1}={\left[a_{11},a_{21},\ldots ,a_{S1}\right]}^{T}$, $f_{2}={\left[a_{12},a_{22},\ldots ,a_{S2}\right]}^{T}$, …, and $f_{F}={\left[a_{1F},a_{2F},\ldots ,a_{SF}\right]}^{T}$. Therefore, the AV Statistical signature is the set $SSAV=\{mean\left(f_{1}\right),std\left(f_{1}\right),max\left(f_{1}\right),min\left(f_{1}\right),P_{25th}\left(f_{1}\right),P_{50th}\left(f_{1}\right),P_{75th}\left(f_{1}\right),mean\left(f_{2}\right),std\left(f_{2}\right),max\left(f_{2}\right),min\left(f_{2}\right),$$P_{25th}\left(f_{2}\right),P_{50th}\left(f_{2}\right),P_{75th}\left(f_{2}\right),\ldots ,mean\left(f_{F}\right),std\left(f_{F}\right),max\left(f_{F}\right),min\left(f_{F}\right),P_{25th}\left(f_{F}\right),P_{50th}\left(f_{F}\right),P_{75th}\left(f_{F}\right)\}$. Then, in the Statistical signature combination step, by domain, combine by row the Statistical signature extracted from each data set [11]. Next (our proposed step: Generalization), by domain, reduce the Statistical signature dimension using the Principal Component Analysis (PCA) [48] technique, so that the resulting number of PCA components [48] is equal for all domains considered and the sum of the variance explained by each PCA component is at least 90%. We use PCA because it is a widely used technique to reduce the dimensions of the feature vectors [27,49], particularly in areas, such as time series prediction [49] (such as domains considered in this document). After that, in the step of reduced Statistical signature labeling, label the reduced Statistical signature (PCA components) of each data set of each domain considered with the best corresponding fusion configuration obtained in the previous stage. Finally, to create the Statistical signature data set, combine by row the tagged and reduced Statistical signature (PCA components) of each data set of each domain considered.

### 3.3. Prediction of the Best Fusion Method

This stage aims to obtain the best fusion strategy for a given data set [11], which can be classified in one of the domains considered. To achieve this goal, by using a k-fold cross-validation strategy [50], train a prediction model (such as RFC), with the Statistical signature data set obtained in the previous stage, to learn to recognize the best fusion configuration for a given data set [11], which fits one of the domains considered. In the next section, we describe the details of how we conduct training and prediction.

## 4. Experimental Methodology and Results

In this section, we present the experiments that we carry out in two steps: (1) To select the best fusion method setting, we make a comparative analysis of different information integration strategies, which were configured in various ways when classifiers, number of characteristics and other variations are used. (2) We train a data-driven prediction model to learn to predict the best fusion strategy of a set of eight for a given data set (not included in the training phase). This data set can be classified in one of the domains considered (SHA, gases, or GFEs). Next, we explain the steps and their main compound (such as the data sets used, the features extracted, and the procedures).

### 4.1. Data Sets

We used 116 original data sets: 40 on SHA, 36 on gases, and 40 on GFEs. Next, we explain the procedures we follow to obtain these original data sets.

#### 4.1.1. SHA Data Sets

The 40 data sets on SHA were extracted from six reference data sets, which researchers commonly use in the recognition of simple human activities [11]. These six data sets used various sensors placed in different parts of a subject’s body, such as accelerometers and gyroscopes. To obtain the group of 40 data sets, we take the possible combinations of two different sensors, say an accelerometer and a gyroscope, which were in the original data sets. So, the difference between each of the 40 resulting data sets was which sensors were included, not their order. In the following, we describe the six original data sets. The University of Texas at Dallas Multimodal Human Action Data set (UTD-MHAD) [51] was collected using a Microsoft Kinect camera and a wearable inertial sensor with a 3-axis accelerometer and a 3-axis gyroscope. This data set includes 27 actions performed by 8 subjects with 4 repetitions for each action. The actions were: 1:swipe left, 2:swipe right, 3:wave, 4:clap, 5:throw, 6:cross arm, 7:basketball shoot, 8:draw X mark, 9:draw circle CW, 10:draw circle CCW, 11:draw triangle, 12:bowling, 13:boxing, 14:baseball swing, 15:tennis swing, 16:arm curl, 17:tennis serve, 18:push, 19:knock, 20:catch, 21:pickup throw, 22:jog, 23:walk, 24:sit to stand, 25:stand to sit, 26:lunge, and 27:squat. As this base has just one gyroscope and one accelerometer (besides the Kinect, which we discarded), it gave us just one data set for the derived data sets collection.The OPPORTUNITY Activity Recognition data set [52] (we call Opportunity) is composed of daily activities recorded with multi modal sensors. It contains recordings of 4 subjects’ activities, including: 1:stand, 2:walk, 3:sit, and 4:lie. It has 2477 instances. Data was recorded by 5 Inertial Measurement Units (IMU) placed in several parts of the subjects’ body: Back (Ba), Right Lower Arm (Rl), Right Upper Arm (Ru), Left Upper Arm (Lu), and Left lower Arm (Ll). The derived 10 data sets from this original one are presented in Table 1.The Physical Activity Monitoring for Aging People Version 2 (PAMAP2) data set [53] was build from the data of three Colibri wireless IMUs: one IMU on the wrist of the dominant arm (Ha), one on the chest (Ch) and one on the dominant side’s ankle (An). Additionally, this data set included data from a Heart Rate monitor: BM-CS5SR from BM innovations GmbH. The data set considers 18 actions performed by nine subjects, including 1:lying, 2:sitting, 3:standing, 4:walking, 5:running, 6:cycling, 7:Nordic walking, 8:TV watching, 9:computer work, 10:car driving, 11:climbing stairs, 12:descending stairs, 13:vacuum cleaning, 14:ironing, 15:laundry folding, 16:house cleaning, 17:soccer playing, and 18:rope jumping. For the derived data sets, we considered the accelerometer and gyroscope data corresponding to the three IMUs for eight actions (1–4, 6, 7, 16, 17) performed by nine subjects, so taking pairs of sensors, we created seven new data sets (see Table 2) (We are not taking all the nine possible combinations due to data set balancing reasons explained later).The Mobile Health (MHealth) data set [54] contains information registered from body motion and vital signs recordings with ten subjects while performing 12 physical activities. The activities are 1:standing still, 2:sitting and relaxing, 3:lying down, 4:walking, 5:climbing stairs, 6:waist bends forward, 7:arms frontal elevation, 8:knees bending (crouching), 9:cycling, 10:jogging, 11:running, and 12:jump front and back. Raw data was collected using three Shimmer2 [55] wearable sensors. The sensors were placed on the subject’s chest (Ch), the right lower arm (Ra) and the left ankle (La), and they were fixed using elastic straps. For data set generation purposes, we took the acceleration and gyro data from the Ra sensor and the La sensor for the first eleven activities, giving us four new data sets (see Table 3).The Daily and Sports Activities (DSA) data set [56] is composed of motion sensor data captured from 19 daily and sports activities, each one carried out by eight subjects during 5 minutes. The sensors used were five Xsens MTx units placed on the Torso (To), Right Arm (Ra), Left Arm (La), Right leg (Rl), and Left leg (Ll). The activities were 1:sitting, 2:standing, 3:lying on back, 4:lying on right side, 5:climbing stairs, 6:descending stairs, 7:standing still in an elevator, 8:moving around in an elevator, 9:walking in a parking lot, 10:walking on a treadmill with a speed of 4 km/h in a flat position, 11:walking on a treadmill with a speed of 4 km/h in a 15 deg inclined position, 12:running on a treadmill with a speed of 8 km/h, 13:exercising on a stepper, 14:exercising on a cross trainer, 15:cycling on an exercise bike in a horizontal position, 16:cycling on an exercise bike in a vertical position, 17:rowing, 18:jumping, and 19:playing basketball. For data set generation purposes, we took some combinations of the accelerometer and gyroscope data corresponding to the five Xsens MTx units, giving us 17 new data sets (see Table 4).The Human Activities and Postural Transitions (HAPT) data set [57] is composed from motion sensor data of 12 daily activities, each one performed by 30 subjects wearing a smartphone (Samsung Galaxy S II) on the waist. The daily activities are 1:walking, 2:walking upstairs, 3:walking downstairs, 4:sitting, 5:standing, 6:laying, 7:stand to sit, 8:sit to stand, 9:sit to lie, 10:lie to sit, 11:stand to lie, and 12:lie to stand. For data set generation, we took the accelerometer and gyroscope to create one new data set.

#### 4.1.2. Gas Data Sets

The 36 derived data sets in the gas domain were extracted from the Gas Sensor Array Drift (GSAD) [13] data set, which is a reference data set often used in gas classification. GSAD collected (during 36 months) data from an array of 16 metal-oxide gas sensors [58] about six different volatile organic compounds: ammonia, acetaldehyde, acetone, ethylene, ethanol, and toluene. Each of these gases was dosed at a wide variety of concentration values ranging from 5 to 1000 ppmv (parts per million by volume). To generate the 36 derived data sets, we formed, using just the information of month 36, different pairs with the 16 sensors (taking into account just the sensors included, not their order). In Table 5, we show the pairs of sensors used to create the derived data sets and the names of those data sets.

#### 4.1.3. GFE Data Sets

The 40 data sets in the GFE domain were extracted from the Grammatical Facial Expressions [14] data set. It is a popular reference data set of the machine learning repository of the University of California, Irvine [59].

The GFE data set was created using a Microsoft Kinect sensor and its purpose is to collect information about emotions from the face expressions. With this information, nine emotion data sets were created and tagged with two classes: Negative (N) or Positive (P). The nine data sets are affirmative (with 541 P and 644 N instances), conditional (with 448 P and 1486 N instances), doubts_question (with 1100 P and 421 N instances), emphasis (with 446 P and 863 N instances), negation (with 568 P and 596 N instances), relative (with 981 P and 1682 N instances), topics (with 510 P and 1789 N instances), Wh_questions (with 643 P and 962 N instances), and yn_questions (with 734 P and 841 N instances).

The information collected by a Kinect sensor consisted of recordings of eighteen videos, and for each frame, one hundred human face point coordinates were extracted and labeled. Coordinates *x* and *y* are width and height dimensions of the face image, and *z* is the depth dimension. The face points correspond to the left eye (0–7), right eye (8–15), left eyebrow (16–25), right eyebrow (26–35), nose (36–47), mouth (48–67), face contour (68–86), left iris (7), right iris (88), nose tip (89), line above left eyebrow (90–94 ), and line above the right eyebrow (95–99).

Using different points from the 100 available on the human face, we created five different groups of these points (see Table 6). Then, we extracted the points of each group to each of the emotion data sets (except the negation one). Therefore, five derived data sets from each of the eight emotion data sets give us a total of 40 original data sets on grammatical facial expression (see Table 7).

Because the number of observations in each class is different in the 40 data sets, we balance the classes in each of them. The imbalance of the classes would cause a classifier to issue results with a bias towards the majority class, especially when using the accuracy metric [60]. The balance strategy we choose is to subsample the majority class [60], which consists of randomly eliminating the observations of the majority class. Specifically, we use a down-sampling implementation for Python: the Scikit-Learn resampling module [61]. This module was configured to resample the majority class without replacement, setting the number of samples to match that of the minority class.

### 4.2. Feature Extraction

We extracted the characteristics of each of the 116 original data sets obtained in the previous Section (40 in the domain of SHA, 36 in the area of gases, and 40 in the field of GFEs). Next, we describe the features extracted by domain.

#### 4.2.1. SHA Features

In the context of human activity, the original signals of a 3-axis accelerometer and a 3-axis gyroscope were segmented into three-second windows (commonly used in this context [62]), without overlapping. Then, 16 features were calculated for each window segment of each of these sensors.

The features we considered for human activity from accelerometers and gyroscopes have been shown to produce good results in this type of applications [63,64]), and they include, for each of the 3 axes, the mean value, the standard deviation, the max value, and the correlation between each pair of axes; likewise, the mean magnitude, the standard deviation of the magnitude, the magnitude area under the curve (AUC, Equation (Equation 1)), and magnitude mean differences between consecutive readings (Equation (Equation 2)). The magnitude of the signal represents the overall contribution of acceleration of the three axes (Equation (Equation 3)).
(1)$$\mathrm{AUC}=\sum_{t=1}^{T}\mathrm{magnitude}\left(t\right),$$
(2)$$\mathrm{meandif}=\frac{1}{T-1}\sum_{t=2}^{T}\mathrm{magnitude}\left(t\right)-\mathrm{magnitude}\left(t-1\right),$$
(3)$$\mathrm{magnitude}\left(x,y,z,t\right)=\sqrt[{}]{a_{x}{\left(t\right)}^{2}+a_{y}{\left(t\right)}^{2}+a_{z}{\left(t\right)}^{2}},$$
where $a_{x}{\left(t\right)}^{2}$, $a_{y}{\left(t\right)}^{2}$, and $a_{z}{\left(t\right)}^{2}$ are the squared accelerations at time interval *t*, and *T* is the last time interval.

#### 4.2.2. Gas Features

In the area of gases, two types of features were extracted: the steady-state characteristic [65] and the exponential moving average ($ema_{\alpha }$) [66]. The steady-state characteristic (considered the standard feature in this context [65]) is defined as the difference of the maximum resistance change and the baseline (Equation (Equation 4)) and its normalized version expressed as the maximum resistance divided by the baseline values (Equation (Equation 5)).
(4)$$\Delta R=\max_{k}r\left[k\right]-\min_{k}r\left[k\right],$$
(5)$$∥\Delta R∥=\frac{\max_{k}r\left[k\right]-\min_{k}r\left[k\right]}{\min_{k}r\left[k\right]}.$$

In these formulas, r[k] is the time outline of the sensor resistance, and *k* is the discrete time that indexes the recording range [0,T] when chemical steam is present in the test chamber.

The exponential moving average ($ema_{\alpha }$) transforms the combined increasing/decreasing discrete time series r[·] of the chemical sensor into a real scalar $f_{\alpha }r[\cdot ]$ by estimating the maximum value (or minimum for the decomposition part of the sensor response) of its $ema_{\alpha }$ (Equation (Equation 6)) [66]. We use this feature because it has been shown to produce good results in previous works [13,66].
(6)y[k]=(1−α)y[k−1]+α(r[k]−r[k−1]).

In Equation (Equation 6), k=1,2,…,T; y[0]=0 (its initial condition); and the scalar α (α=0.1, α=0.01, and α=0.001) is an operator smoothing parameter that defines both the quality of the characteristic $f_{\alpha }r[\cdot ]$ and the moment of its appearance throughout the series temporary. From these values α, three different characteristic values are obtained from the pre-recorded ascending part of the sensor response. In addition, three additional characteristics are obtained with the same values α for the decaying part of the sensor response.

#### 4.2.3. GFE Features

In the GFE field, three types of features were obtained from the data collected by the Microsoft Kinect sensor. Although, in this case, we only take data from one sensor, we take advantage of the multi-view learning paradigm [67] by extracting different sets of features or views in the data. This paradigm, which addresses the problem of learning a model based on the various aspects of the data [67], has shown a great practical success [68]. In our case, the views are the entries to our merge configuration.

Those three types of characteristics (already used with good results in this field [14]) are the distances, angles, and depth dimensions. For each GFE data set containing facial points (see Section 4.1.3) per frame, six distances were extracted from 12 of these facial points, three angles were obtained from nine facial points, and 12 depth dimensions were taken from the 12 points used to calculate the distances. Then, these 21 features extracted per frame (six distances, three angles, and 12 depth dimensions) are concatenated according to the size of the overlapping window (overlapped by one frame) defined by the type of emotion. The sizes of these windows are 2 for Affirmative, 3 for Conditional, 5 for Doubts, 2 for Emphasis, 6 for Relative, 4 for Thematic, 6 for Wh-questions, and 2 Yes/no questions. We use these window sizes because they have shown that they produce good results in this area [14].

### 4.3. Finding the Best Configuration of the Fusion Methods

We describe, in five steps, the procedure we follow to find, for each data set described in Section 4.1, the best of the configurations of the fusion methods defined in Section 3.1. We wrote Python code on the Jupyter Notebook platform [69] to create functions that implement the fusion strategy settings studied here. These functions used the cross-validation technique with three folds.For each of the data sets considered here, we obtained 24 accuracy samples for each configuration of the fusion strategies. We got these samples by executing, 24 times, each of the functions created in the previous step. In cases where a fusion configuration shuffles characteristics, each of its 24 accuracies was obtained by taking the best accuracy resulting from executing the function that implements it 33 times. At each run, a different combination of features was attempted. We executed this function 33 times to try to find the best accuracy that this fusion configuration can achieve.We performed the Friedman test [42] with the data obtained in step 2 to see if there are significant differences (with a confidence level of 95%) between some pairs of these configurations.We performed the Holm test [43] with the data obtained in step 3 to see if there is a significant difference (with a confidence level of 95%) between the Aggregation (considered here the baseline for comparison purposes) and some other configuration.We identify the best configuration of the fusion methods for each data set considered here, based on the results of the previous step. Specifically, for each data set, from the configurations that presented a statistically significant difference (according to the results of the Holm test of the previous step), we identified the one that achieved the greatest of these differences. We consider this configuration to be the best. In Table 8, Table 9 and Table 10, we present the best configurations of the fusion methods for each of the data sets considered here.

### 4.4. Recognition of the Best Configuration of the Fusion Strategies

In this section, we present the main steps that we follow to recognize the best configuration of the fusion strategies for a given data set, which can be classified in one of the domains considered here (SHA, gas detection, or GFEs). Next, we present each of these steps.We created the Statistical signature data set following the procedure defined in Stage Section 3.2. First, we extracted for each of the characteristics (described in Section 4.2) of the data sets (presented in Section 4.1) their Statistical signatures (defined in Section 3.2), which are the mean, the standard deviation, the maximum value, and the minimum value, i.e., the 25th, 50th, and 75th percentiles.Then, we created three sets of meta-data, one for each domain considered here (SHA, Gas, and GFE), where each row of each one of them corresponds to the Statistical signature of each of the data sets of the corresponding domain.Next, we reduced to 15 the number of columns of each of these three meta-data sets using the PCA [48] technique, for a dimensional reduction of the digital signature of the data sets of the corresponding domain. We took this number because we obtained the same amount of PCA compounds [48] per domain, and the sum of the explained variance of the corresponding compounds per domain was at least 90% (as indicated in Section 3.2): 98% for the SHA domain database, 99% for the Gas domain database, and 92% for the GFE domain database.After that, each row of each of these three meta-data sets was labeled with the MultiviewStacking, MultiViewStackingNotShuffle, Voting, or AdaBoost tags, extracted from the results of Table 8, Table 9 and Table 10. We chose the results from these tables because they show the best configurations of merge strategies, for each data set described in Section 4.1. They are the best configurations since they present the greatest significant differences concerning the Aggregation configuration. In cases where there were no significant differences between these configurations and the Aggregation configuration, we took the latter as the best option. Therefore, for the data sets that are in these cases, we tagged them with the Aggregation string.Finally, we combine, by row, the three meta-data sets, forming our Statistical signature data set (the generalized meta-data set). We present the final dimension and class distribution of the Statistical signature data set in Table 11.We balanced the Statistical signature data set because, in this data set, the number of observations in each class was different (class imbalance condition). This circumstance would result in a classifier issuing results with a bias towards the majority class. Even though there are different approaches to address the problem of class imbalance [60], we chose the up-sampling strategy. This strategy increases the number of minority class samples by using multiple instances of minority class samples. In particular, we used an up-sampling implementation for Python: the Scikit-Learn resampling module [61]. This module was configured to resample the minority class with replacement so that the number of samples for this class matches that of the majority class. In Table 12, we present the new dimension and evenly class distributions of the Statistical signature data set.To learn to recognize the best fusion configuration for a given data set that belongs to one of the domains considered here, using the data from Statistical signature data set, we trained and validated the RFC classifier using a three-fold cross-validation strategy [50]. We measure the performance of this classifier in terms of accuracy, precision, recall, f1-score [70], and support. We summarize the performance of this classifier in Table 13 and Table 14.

### 4.5. Experimental Results and Discussion

In Table 8, Table 9 and Table 10, we present the best of the configurations of the fusion methods (defined in Section 3.1) for each of the data sets described in Section 4.1. In these tables, for each data set, we mark with a checkmark the best configuration that achieved the greatest statistical significant difference among the configurations of the fusion methods that also achieved a significant statistical difference, with a confidence level of 95%. This is done by comparing the accuracies achieved by such configurations and the accuracies achieved by the Aggregation configuration.

In the following analysis, we say that a configuration of a fusion method is “marked” for a given data set, provided it is the best.

In Table 8, we can notice that multi-view stacking with shuffled features is marked in 14 of 40 SHA data sets; multi-view stacking is marked in 11 of 40 SHA data sets; and Adaboost is marked only for the OpportunityBaAccLuGy data set. These observations suggest that the marked data sets have some properties that favor the corresponding fusion method configuration. The results are consistent with findings presented by Aguileta et al. [11], where they compared the same fusion configuration using similar SHA data sets.

In Table 9, we see that only multi-view stacking with shuffled features is marked in 17 of 36 Gas data sets. In Table 10, we notice that Voting with shuffled features is marked in 17 of 40 GFE data sets; multi-view stacking with shuffled features is marked in 6 of 40 GFE data sets; and Multi-view stacking is marked in 3 out of 40 GFE data sets. Once again, the observations suggest that the marked data sets have some properties that favor the corresponding fusion method configuration.

The results of Table 8, Table 9 and Table 10 are in line with findings reported in Reference [11] in the sense that there is no method to merge sensor information that is better independent of the situation and types of sensors, as it can be argued in the following conclusions. The multi-view stacking configuration was the best for GAS data sets collected with gas sensors; Voting achieves the best accuracy, followed by Multi-view stacking with its two configurations (shuffled characteristics and unshuffled characteristics, respectively), for GFE data sets collected with the Kinect sensor; and Multi-views stacking with its two settings, followed by AdaBoost, were the best ones for SHA data sets collected by the accelerometer sensor and gyroscope sensor.

In Table 11, we can see some key characteristics from the Statistical signature data set, step 1 of Section 4.4, such as its dimensions and the distribution of classes. The dimensions are 116 rows that correspond to the Statistical signature of each of the 116 data sets of the three domains considered here (see Section 4.1) and 16 columns that correspond to 15 PCA compounds (see step 1 of Section 4.4) and the label. We highlight that, since we work with a similar number of samples per domain (40 in SHA, 36 in gases and 40 in GFE for a total of 116 data sets), we can consider that this Statistical signature data set is balanced from the perspective of the number of samples per domain. In this work, we are interested in balancing samples by domain to avoid possible bias towards a domain during the prediction of the best fusion strategy. About the distribution of classes, we notice an imbalance of classes.

The result of balancing the classes of the Statistical signature data set is shown in Table 12, step 2 of Section 4.4.

The results of identifying the best configuration of the fusion strategies are presented in Table 13 and Table 14, corresponding to step 3 of Section 4.4.

In Table 13, we observe that Voting was the method that RFC predicted with the highest number of hits (46/47), followed by MultivewStackingNotShuffle (with 43/47 hits), Aggregation (with 41/47 hits), and MultivewStacking (with 38/47 hits). These remarks suggest that RFC, when trained with our proposed Statistical signature data set, can predict Voting well and can predict MultiviewStackingNotShuffle, Aggregation, and MultiviewStacking reasonably well.

In Table 14, it is clear that the precision metric and f1-score metric have their highest value with Voting, followed by MultiviewStackingNotShuffle, Aggregation, and MultiviewStacking. These observations confirm that the RFC classifier, when trained with our Statistical signature data set, can predict Voting very well and can predict MultiviewStackingNotShuffle, Aggregation, and MultiviewStacking reasonably well.

In Table 14, we can also notice that, on average, the four metrics (accuracy, precision, recall, and f1-score) have the same value of 91%. Therefore, based on the averaged value reached by the metrics, we support our affirmation of a good prediction of the best data fusion method, among the five candidates, for a given data set, which can be classified into one of the three types of information considered here (SHA, Gas, or GFE). These candidates were among the best in our comparative analysis of fusion methods when trying to predict, in terms of accuracy, the classes in the data set (described in Section 4.1) of the three domains. This result represents, in our opinion, a breakthrough in the topic of sensor information fusion.

## 5. Conclusions

In this article, we presented a method aimed to predict the best strategy to merge data from different types of sensors, eventually belonging to completely different domains (SHA, gas types, and GFE). We claim that EPOFM is a general method for choosing the best fusion strategy from a set of given ones.

Our contribution is that we proposed and experimentally validated a completely new approach for finding the best fusion method, in which we construct a meta-data set where each row corresponds to the Statistical signature of one source data set, and then we train a supervised classifier with this meta-data set to predict the best fusion method. To the best of our knowledge, this approach has never been proposed before, and for good reason: when we map a whole source data set to only one row of the meta-data set, then, in order to train a classifier, we need many rows—so, many source data sets; in our experiments, this is in the order of the hundreds.

Further, when extending in this paper our previous work (POFM), which was restricted to activity recognition, to a general method (EPOFM), the latter becomes able to merge in a single meta-data set the Statistical signatures of completely different source domains. In the EPOFM method, based on the POFM approach, we modified the Statistical signature meta-data set (defined in the POFM method), adding a Generalization step, in which the features of each domain were normalized to fs features (where fs is a parameter) by using PCA or equivalent method, so that independently of the domain, the selection of the best method could be done independently of the source domains. The generality of our method is illustrated in this article using data sets as diverse as gas types and facial expressions, in addition to the human activity used in the original POFM method. The fact that we were able to predict with high accuracy the best fusion strategy in all the considered different domains provides objective evidence supporting the generality claim of this method.

Our results confirmed the observation of Aguileta et al. [11] that there is not a single best method to merge sensor information independently of the situation and the types of sensors, so, in most cases, a choice has to be made among different fusion strategies, but now this observation has been shown to remain across other very different domains from human activity recognition.

We presented experimental results showing that our generalized approach could predict, with an accuracy of 91%, the best of five strategies (Voting that shuffles features, Aggregation, Multi-view stacking that shuffles features, Multi-view stacking not shuffling features, and AdaBoost). These five fusion methods were selected from eight candidate fusion methods (Aggregation, Multi-view stacking (with three configurations), Voting (with three arrangements) and AdaBoost) because they achieved the highest accuracy when classes were predicted, using a prediction model, of 40 SHA data sets, 36 gas data sets, and 40 GFE data sets considered in this work. The reported 91% accuracy prediction was achieved (as defined in the POFM method) by a machine learning method, using the statistic signatures of the features extracted from 116 data sets.

Finally, we discuss the practical value of our approach. At this point, we could not claim that using our method is always better (in terms of effort or time) than merely trying several configurations directly over the data set by hand. Nevertheless, we point out two practical reasons for using it: first, it provides a systematic way of having a “default” configuration, just by computing for the given data set its Statistical signature and then getting a prediction from our already trained system, being these are straightforward processes. This default configuration can be used only as an initial reference, to be applied or not, according to the best judgment of the person involved in the process. Second, there could be some very specific scenarios where the application of a method like ours is almost mandatory, like, for instance, in real-time applications where some sensors may fail or new ones are incorporated. In this situation, our approach would find in real-time the best configuration for the new situation, without the need for any training. Considering that trying out and evaluating every possible configuration does require training, it is clear that a real-time response would not be possible. However, beyond very specific cases like the ones mentioned above, the practical utility of using a general configuration predictor as ours has to be judged by a well-informed decision-maker; a quantitative framework for making such a decision is well outside the scope of this paper.

### Future Work

In the short term, we are going to make the Statistical signature data set public.

We consider that it could be possible to improve the composition of the data set of Statistical signatures, as our research in this aspect has been limited, and perhaps the overall accuracy could be further improved. In addition, we could test additional fusion strategies and data sets for further generality.

On the other hand, we could experiment with the imbalance property, from the perspective of the number of samples per domain, of the Statistical signature data set. A Statistical signature data set with this property could be more flexible during its construction process since it would allow increasing the examples of any domain on demand. Augmenting such samples on demand would speed up the use of our approach (which predicts the best fusion method) since you will not have to wait for a balanced number of samples from a given domain to be included in the Statistical signature data set.

Furthermore, in the future, we could try to predict the best fusion method for data sets from different domains than those considered in the Statistical signature data set, in a transfer learning style.

## Figures and Tables

**Figure 1 sensors-20-02350-f001:**
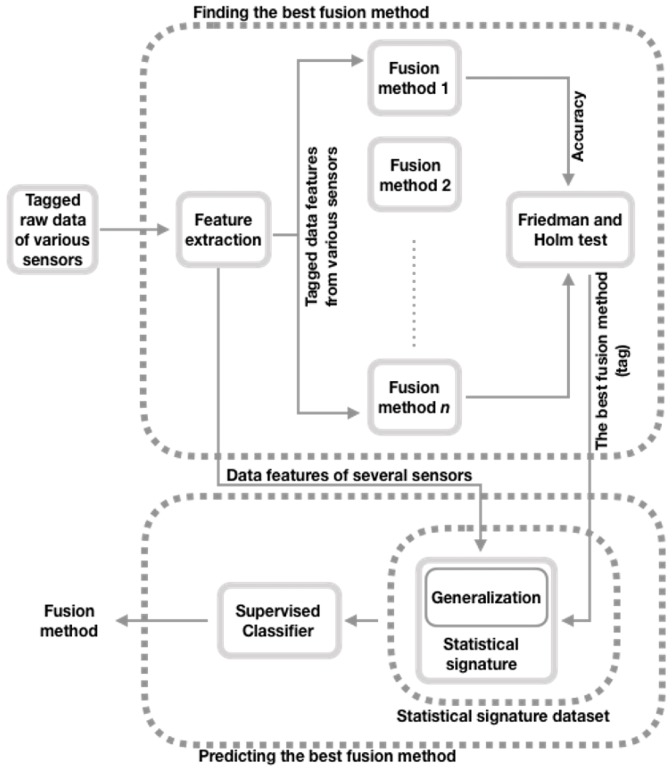
Overview of the extended method that predicts the optimal fusion method.

**Figure 2 sensors-20-02350-f002:**
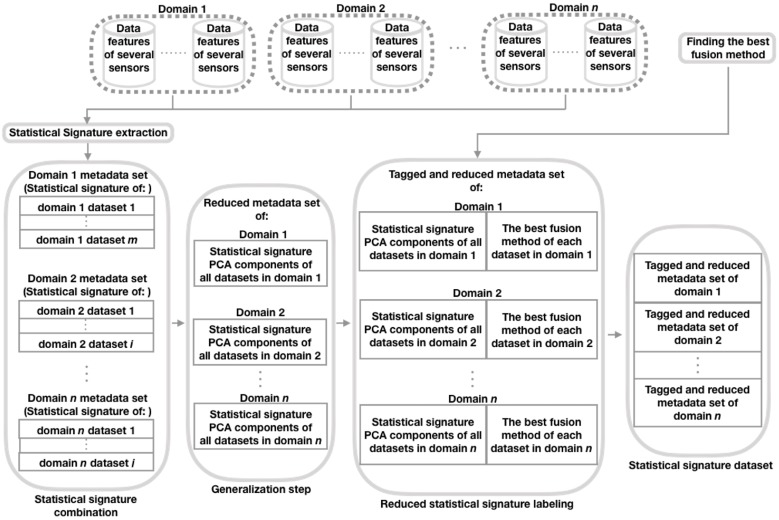
Procedure to create the Statistical signature data set. PCA = Principal Component Analysis.

**Table 1 sensors-20-02350-t001:** Data sets obtained from the data of the Inertial Measurement Units (IMUs) of the Opportunity data set.

Name	Sensors
OpportunityRlAccGy	Accelerometer and Gyroscope of the Rl
OpportunityBaAccLlGy	Ba Accelerometer and Ll Gyroscope
OpportunityBaAccLuGy	Ba Accelerometer and Lu Gyroscope
OpportunityBaAccRlGy	Ba Accelerometer and Rl Gyroscope
OpportunityBaAccRuGy	Ba Accelerometer and Ru Gyroscope
OpportunityLlAccBaGy	Ll Accelerometer and Ba Gyroscope
OpportunityLlAccGy	Accelerometer and Gyroscope of the Ll
OpportunityRuAccGy	Accelerometer and Gyroscope of the Ru
OpportunityRuAccLlGy	Ru Accelerometer and Ll Gyroscope
OpportunityRuAccLuGy	Ru Accelerometer and Lu Gyroscope

**Table 2 sensors-20-02350-t002:** Data sets obtained from the data of the IMUs of the PAMAP2 data set.

Name	Sensors
PAMAP2HaAccGy	Accelerometer and Gyroscope of the Ha
PAMAP2AnAccGy	Accelerometer and Gyroscope of the An
PAMAP2AnAccHaGy	An Accelerometer and Ha Gyroscope
PAMAP2ChAccGy	Accelerometer and Gyroscope of the Ch
PAMAP2ChAccHaGy	Ch Accelerometer and Ha Gyroscope
PAMAP2HaAccAnGy	Ha Accelerometer and An Gyroscope
PAMAP2HaAccChGy	Ha Accelerometer and Ch Gyroscope

**Table 3 sensors-20-02350-t003:** Data sets obtained from the data of the IMUs of the Mhealth data set.

Name	Sensors
MHealthRaAccGy	Accelerometer and Gyroscope of the Ra
MHealthLaAccGy	Accelerometer and Gyroscope of the La
MHealthLaAccRaGy	La Accelerometer and Ra Gyroscope
MHealthRaAccLaGy	Ra Accelerometer and La Gyroscope

**Table 4 sensors-20-02350-t004:** Data sets obtained from the data of the IMUs of the DSA data set.

Name	Sensors
DSALaAccLlGy	La Accelerometer and Ll Gyroscope
DSALaAccRlGy	La Accelerometer and Rl Gyroscope
DSALlAccLaGy	Ll Accelerometer and La Gyroscope
DSALlAccRaGy	Ll Accelerometer and Ra Gyroscope
DSALlAccRlGy	Ll Accelerometer and Rl Gyroscope
DSARaAccRlGy	Ra Accelerometer and Rl Gyroscope
DSARlAccLaGy	Rl Accelerometer and La Gyroscope
DSARlAccLlGy	Rl Accelerometer and Ll Gyroscope
DSARlAccRaGy	Rl Accelerometer and Ra Gyroscope
DSARlAccToGy	Rl Accelerometer and To Gyroscope
DSARaAccGy	Accelerometer and Gyroscope of the Ra
DSALaAccGy	Accelerometer and Gyroscope of the La
DSALlAccGy	Accelerometer and Gyroscope of the Ll
DSARlAccGy	Accelerometer and Gyroscope of the Rl
DSAToAccGy	Accelerometer and Gyroscope of the To
DSAToAccLlGy	To Accelerometer and Ll Gyroscope
DSAToAccRaGy	To Accelerometer and Ra Gyroscope

**Table 5 sensors-20-02350-t005:** Data sets obtained from gas sensor pairs from GSAD data set for month 36.

Name	Sensors
S7S8gas	Gas sensors 7 and 8
S6S16gas	Gas sensors 6 and 16
S12S15gas	Gas sensors 12 and 15
S10S15gas	Gas sensors 10 and 15
S5S15gas	Gas sensors 5 and 15
S1S2gas	Gas sensors 1 and 2
S3S16gas	Gas sensors 3 and 16
S9S15gas	Gas sensors 9 and 15
S2S15gas	Gas sensors 2 and 15
S13S15gas	Gas sensors 13 and 15
S8S15gas	Gas sensors 8 and 15
S3S15gas	Gas sensors 3 and 15
S13S16gas	Gas sensors 13 and 16
S4S16gas	Gas sensors 4 and 16
S5S6gas	Gas sensors 5 and 6
S10S16gas	Gas sensors 10 and 16
S11S16gas	Gas sensors 11 and 16
S1S16gas	Gas sensors 1 and 16
S7S16gas	Gas sensors 7 and 16
S8S16gas	Gas sensors 8 and 16
S11S15gas	Gas sensors 11 and 15
S9S10gas	Gas sensors 9 and 10
S11S12gas	Gas sensors 11 and 12
S14S15gas	Gas sensors 14 and 15
S13S14gas	Gas sensors 13 and 14
S1S15gas	Gas sensors 1 and 15
S4S15gas	Gas sensors 4 and 15
S3S4gas	Gas sensors 3 and 4
S5S16gas	Gas sensors 5 and 16
S14S16gas	Gas sensors 14 and 16
S2S16gas	Gas sensors 2 and 16
S15S16gas	Gas sensors 15 and 16
S12S16gas	Gas sensors 12 and 16
S7S15gas	Gas sensors 7 and 15
S9S16gas	Gas sensors 9 and 16
S6S15gas	Gas sensors 6 and 15

**Table 6 sensors-20-02350-t006:** Groups created with facial points.

Group Name	Facial Points
V1	17, 27, 10, 89, 2, 39, 57, 51, 48, 54, 12
V2	16, 36, 1, 41, 9, 42, 69, 40, 43, 85, 50, 75, 25, 37, 21, 72, 58, 48, 77, 54
V3	95, 31, 96, 32, 88, 14, 11, 13, 61, 67, 51, 58, 97, 98, 27, 10, 12, 15, 62, 83, 66
V4	91, 3, 18, 73, 69, 39, 42, 44, 49, 59, 56, 86, 90, 68, 6, 70, 63, 80, 78
V5	24, 32, 46, 28, 33, 80, 39, 44, 61, 63, 59, 55, 92, 20, 23, 74, 41, 49, 89, 53

**Table 7 sensors-20-02350-t007:** Data sets obtained from the facial points of the five groups created (V1–V5).

Name of Data Sets Created with Group Points:				
V1	V2	V3	V4	V5
affirmativeV1	affirmativeV2	affirmativeV3	affirmativeV4	affirmativeV5
conditionalV1	conditionalV2	conditionalV3	conditionalV4	conditionalV5
doubts_questionV1	doubts_questionV2	doubts_questionV3	doubts_questionV4	doubts_questionV5
emphasisV1	emphasisV2	emphasisV3	emphasisV4	emphasisV5
relativeV1	relativeV2	relativeV3	relativeV4	relativeV5
topicsV1	topicsV2	topicsV3	topicsV4	topicsV5
Wh_questionsV1	Wh_questionsV2	Wh_questionsV3	Wh_questionsV4	Wh_questionsV5
yn_questionsV1	yn_questionsV2	yn_questionsV3	yn_questionsV4	yn_questionsV5

**Table 8 sensors-20-02350-t008:** Best fusion method for each simple human activities (SHA) data set. A tick (✔) marks the best configuration when it is statistical-significantly better than aggregation; otherwise, it is left blank.

SHA	Voting (Shuffled Features)	Voting	Voting All Features CART-LR-RFC	Multi-View Stacking (Shuffle)	Multi-View Stacking	Multi-View Stacking All Features CART-LR-RFC	AdaBoost
DSARlAccRaGy					✔		
PAMAP2HaAccGy							
OpportunityLlAccGy							
PAMAP2HaAccAnGy							
OpportunityRlAccGy							
DSALaAccRlGy					✔		
DSALlAccLaGy					✔		
DSALlAccRaGy					✔		
OpportunityRuAccLuGy				✔			
DSARlAccToGy					✔		
DSALlAccRlGy				✔			
DSARaAccRlGy					✔		
DSARlAccLlGy					✔		
DSALaAccGy							
HAPT				✔			
DSALlAccGy				✔			
MHealthLaAccRaGy				✔			
DSARaAccGy				✔			
OpportunityBaAccLuGy							✔
OpportunityRuAccLlGy							
MHealthRaAccLaGy				✔			
OpportunityLlAccBaGy							
DSARlAccGy							
MHealthRaAccGy				✔			
DSALaAccLlGy					✔		
DSAToAccRaGy				✔			
OpportunityBaAccLlGy							
OpportunityBaAccRlGy				✔			
PAMAP2ChAccHaGy							
OpportunityBaAccRuGy				✔			
PAMAP2AnAccHaGy					✔		
OpportunityRuAccGy							
PAMAP2ChAccGy							
DSAToAccLlGy				✔			
MHealthLaAccGy				✔			
PAMAP2HaAccChGy					✔		
PAMAP2AnAccGy							
UTD-MHAD							
DSARlAccLaGy					✔		
DSAToAccGy				✔			

**Table 9 sensors-20-02350-t009:** Best fusion method for each Gas data set. A tick (✔) marks the best configuration when it is statistical-significantly better than aggregation.

Gas	Voting (Shuffled Features)	Voting	Voting All Features CART-LR-RFC	Multi-View Stacking (Shuffle)	Multi-View Stacking	Multi-View Stacking All Features CART-LR-RFC	AdaBoost
S7S8gas				✔			
S6S16gas							
S12S15gas							
S10S15gas							
S5S15gas							
S1S2gas							
S3S16gas							
S9S15gas							
S2S15gas							
S13S15gas				✔			
S8S15gas				✔			
S3S15gas				✔			
S13S16gas				✔			
S4S16gas				✔			
S5S6gas							
S10S16gas							
S11S16gas				✔			
S1S16gas							
S7S16gas							
S8S16gas				✔			
S11S15gas				✔			
S9S10gas				✔			
S11S12gas							
S14S15gas				✔			
S13S14gas							
S1S15gas							
S4S15gas				✔			
S3S4gas							
S5S16gas				✔			
S14S16gas				✔			
S2S16gas							
S15S16gas				✔			
S12S16gas				✔			
S7S15gas				✔			
S9S16gas							
S6S15gas							

**Table 10 sensors-20-02350-t010:** Best fusion method for each GFE data set. A tick (✔) marks the best configuration when it is statistical-significantly better than aggregation. RFC = Random Forest; LR = Logistic Regression; CART = Decision Tree.

GFE	Voting (Shuffled Features)	Voting	Voting All Features CART-LR-RFC	Multi-View Stacking (Shuffle)	Multi-View Stacking	Multi-View Stacking All Features CART-LR-RFC	AdaBoost
emphasisV4							
yn_questionV1							
wh_questionV3	✔						
wh_questionV2	✔						
yn_questionV5							
doubt_questionV1	✔						
wh_questionV5	✔						
emphasisV3							
conditionalV5					✔		
conditionalV1				✔			
emphasisV1							
conditionalV3	✔						
emphasisV5							
relativeV3	✔						
topicsV4	✔						
topicsV5	✔						
doubt_questionV5					✔		
wh_questionV1	✔						
affirmativeV3							
yn_questionV3							
topicsV2	✔						
doubt_questionV2	✔						
emphasisV2							
doubt_questionV3				✔			
relativeV5				✔			
yn_questionV4							
relativeV2	✔						
topicsV3	✔						
topicsV1	✔						
doubt_questionV4	✔						
relativeV1				✔			
affirmativeV5	✔						
yn_questionV2							
affirmativeV1							
wh_questionV4	✔						
conditionalV4					✔		
affirmativeV2							
relativeV4				✔			
conditionalV2				✔			
affirmativeV4							

**Table 11 sensors-20-02350-t011:** Dimensions and class distribution of the Statistical signature data set.

		Class Distribution				
Dataset	Dimensions (Rows, Columns)	Aggregation	Multiview Stacking	Voting	Multiview Stacking NotShuffle	Adaboost
Statistical signature	(116, 16)	47	37	17	14	1

**Table 12 sensors-20-02350-t012:** Balanced Statistical signature data set.

		Class Distribution				
Dataset	Dimensions (Rows, Columns)	Aggregation	Multiview Stacking	Voting	Multiview Stacking NotShuffle	Adaboost
Statistical signature	(235, 16)	47	47	47	47	47

**Table 13 sensors-20-02350-t013:** Confusion matrix of RFC based on the Statistical signature data set.

Label	Adaboost	Aggregation	Multiview Stacking	Multiview Stacking NotShuffle	Voting
Adaboost	47	0	0	0	0
Aggregation	0	41	4	2	0
MultiViewStacking	0	5	38	3	1
MultiViewStackingNotShuffle	0	1	0	43	3
Voting	0	0	1	0	46

**Table 14 sensors-20-02350-t014:** Performance metrics of RFC based on the Statistical signature data set.

Label	Precision	Recall	f1-Score	Support
Adaboost	1.00	1.00	1.00	47
Aggregation	0.87	0.87	0.87	47
MultiViewStacking	0.88	0.81	0.84	47
MultiViewStackingNotShuffle	0.90	0.91	0.91	47
Voting	0.92	0.98	0.95	47
avg/total	0.91	0.91	0.91	235
accuracy			0.91	235

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
