# Peer review of "Choosing the Best Sensor Fusion Method: A Machine-Learning Approach"

_sensors, 2020, doi:10.3390/s20082350_

Round 1

Reviewer 1 Report

  1. The name: “Fingerprint Data Set” was a little confusing for me, because it suggests data used in the biometric field. Please add some explanation of this at the start of the subsection 3.2 (e.g. repeating the information from lines 44-54) or please change the name (depends on your previous works).
  2. The way of creating the statistical signature’should be described with more details.
  3. Why did you use only 36 data sets of Gas Data sets (Tab 2)? You have data with 16 sensors so you are able to create 120 data sets. What about dataset S1S3gasDataset10? Why didn’t you use it?
  4. You should explain with details the names of data set used. What does mean e.g. ‘dailyRIAccRaGy’? – DailySports data set with sensors placed on the right leg, right arm and …..?
  5. Please add more detailed discussion of the results.
  6. You should to reduce the font size in the Tables 4-7

Author Response

Answer to reviewer remarks 1

Remarks of the reviewer 1

1.The name: “Fingerprint Data Set” was a little confusing for me, because it suggests data used in the biometric field. Please add some explanation of this at the start of the subsection 3.2 (e.g. repeating the information from lines 44-54) or please change the name (depends on your previous works).

Answer to this remark by authors:

We have changed the name "Fingerprint DataSet" to "Statistical Signature DataSet" (see lines: 45, 53,  103, 172, 200, 205, 388, 409, 411, 412,  420, 423, 455, 460, 465, 472, 476, 502, 529, 531, 534, 538,and 540. Also, see figures 1 and 2. Besides, see tables 11, 12, 13, and 14.

2. The way of creating the statistical signature’ should be described with more details.

Answer to this remark by authors:

We now explain the details of the process to create the statistical signatures. Then, we present an example in mathematical notation (see lines 181-190).

3. Why did you use only 36 data sets of Gas Data sets (Tab 2)? You have data with 16 sensors so you are able to create 120 data sets. What about dataset S1S3gasDataset10? Why didn’t you use it?

Answer to this remark by authors:

We create only 36 Gas datasets because we are interested in having the most balanced Statistical signature dataset possible in terms of domains (40 datasets on HAR and 40 on GFE). Each row in this data set corresponds to the statistical signature of each data set in each domain. It is not known what consequences a major domain imbalance could have. An unbalanced Statistical signature dataset, as discussed here, is subject to further investigation. The above is discussed on lines 459-463 and 533-538.

As for the S1S3gasDataset10 dataset, it was not necessary as we already have the 36 datasets that we needed.

4. You should explain with details the names of data set used. What does mean e.g. ‘dailyRIAccRaGy’? – DailySports data set with sensors placed on the right leg, right arm and …..? 

Answer to this remark by authors:

We explain the details of the names of the data sets used in Tables 1- 7.

5. Please add more detailed discussion of the results.

Answer to this remark by authors:

Discussions are presented as results are shown. The name of the Experimental Results section is changed to Experimental Results and Discussion to clarify this fact. Additionally, we add more discussion on lines 459-463, 523-527, and 533--540.

6. You should to reduce the font size in the Tables 4-7

Answer to this remark by authors:

We reduced the fonts in the suggested tables. We inserted other tables in the paper, so these tables are now tables 11-14

Reviewer 2 Report

The paper can be accepted if the following minor amendments are addressed:

Recommendations are missing to say for which type of application which fusion method should be selected.

The Friedman’s rank test and Holm’s test should be described, as they are not very commonly used and therefore it is required to explain to the reader.

Figures 1 and 2 are not really explained in the text. Especially Figure 2 stands on his own.

Section 4.1.1 (2): it would be interesting to know more in detail where the sensors on the human body were placed.

Author Response

Answer to reviewer remarks 2

Remarks of the reviewer 2

The paper can be accepted if the following minor amendments are addressed:

1.Recommendations are missing to say for which type of application which fusion method should be selected.

Answer to this remark by authors:

Suggesting the type of application by the fusion method is beyond the scope of this research. This work was focused on domains and not applications. Specifically, we present a general approach to learning how to predict the best method of merging any dataset from any domain, as long as this domain is included in the statistical signature dataset.

2.The Friedman’s rank test and Holm’s test should be described, as they are not very commonly used and therefore it is required to explain to the reader.

Answer to this remark by authors

We use Friedman's rank test because it has been shown to be safer and stronger than parametric tests, such as ANOVA, in the context of comparing classifiers across multiple data sets [1]. Also, the Holm simple test is preferable to more complex tests, such as the Hommel post hoc test, since both tests produce similar results [1]. The above is discussed in lines 164-171. 

Besides, Friedman's rank test and Holm's test are part of the classic statistics body of knowledge [2]. 

Reference: 

[1] Demšar, J. Statistical comparisons of classifiers over multiple data sets. Journal of Machine learning research 2006, 7, 1–30.

[2] Hollander, M.; Wolfe, D. A. (1973). Nonparametric Statistics. New York: J. Wiley. ISBN 978-0-471-40635-8.

3.Figures 1 and 2 are not really explained in the text. Especially Figure 2 stands on his own.

Answer to this remark by authors:

We refer to Figure 1 in lines 100 and 106. Lines 100-208 explain this figure. Additionally, we refer to Figure 2 in lines 177 and 180. Lines 173-201 describe Figure 2.

4.Section 4.1.1 (2): it would be interesting to know more in detail where the sensors on the human body were placed.

Answer to this remark by authors:

We explain the detail of where the sensors were placed in the human body for each data set in Tables 1-4.

Reviewer 3 Report

The paper proposes a machine-learning data-driven approach to select the best data fusion approach in a given domain.

The idea of the authors is quite novel (besides the paper being an extension of a previously published article that was focused on a specific application domain) and appears generalized.

However the main and major concern is the practical impact of the proposed approach. In reviewer's understanding, the researcher while analyzing - let us say - emotion detection by means of multiple sensor sources (e.g. physiological, audio-video, inertial signals) would first generate a significant data set from which multiple sub-data sets should be generated and/or a number of other, independent open data sets should be found (the former can be very time-consuming while the latter is not always a viable option). Then, the proposed method would be able to identify the best fusion strategy. 

This step would be required even in case some literature (based on the proposed approach) would suggests a convergence on a specific fusion approach in the same "macro domain" (emotion detection in our example).

Therefore, it is not clear whether there is a practical advantage of this approach (e.g. in terms of research/analysis efforts) as opposed to "simply" evaluate different fusion techniques directly over a "reasonably smaller" dataset. Trying to clarify and possibly quantitatively evaluate pros and cons of these alternatives is critical to convince a possible reader that the time (but might not be the only metric of the effort) required for the datasets generation is a wise investment. 

Finally, in the conclusion section, the authors could highlight ongoing and future works in this research line.

Author Response

Answer to reviewer remarks 3

Remarks of the reviewer 3

The paper proposes a machine-learning data-driven approach to select the best data fusion approach in a given domain.

1. The idea of the authors is quite novel (besides the paper being an extension of a previously published article that was focused on a specific application domain) and appears generalized.

However the main and major concern is the practical impact of the proposed approach. In reviewer's understanding, the researcher while analyzing - let us say - emotion detection by means of multiple sensor sources (e.g. physiological, audio-video, inertial signals) would first generate a significant data set from which multiple sub-data sets should be generated and/or a number of other, independent open data sets should be found (the former can be very time-consuming while the latter is not always a viable option). Then, the proposed method would be able to identify the best fusion strategy. 

This step would be required even in case some literature (based on the proposed approach) would suggests a convergence on a specific fusion approach in the same "macro domain" (emotion detection in our example).

Therefore, it is not clear whether there is a practical advantage of this approach (e.g. in terms of research/analysis efforts) as opposed to "simply" evaluate different fusion techniques directly over a "reasonably smaller" dataset. Trying to clarify and possibly quantitatively evaluate pros and cons of these alternatives is critical to convince a possible reader that the time (but might not be the only metric of the effort) required for the datasets generation is a wise investment. 

Answer to this remark by authors:

The practical concerns mentioned by the reviewer are very relevant indeed. First, we do not claim that the method we propose is the most practical in every situation. But having a method like the one we proposed could be valuable in applications where changes in sensor availability could make another setting (merge method) the most appropriate at the time. Our model could provide an immediate response to this scenario, as opposed to having to test each merge method against the source data and having to choose the best one manually. The above is discussed in lines 523-527.

2. Finally, in the conclusion section, the authors could highlight ongoing and future works in this research line.

Answer to this remark by authors:

We discuss future additional work in lines 533-540.

Round 2

Reviewer 3 Report

I appreciate the efforts of the authors trying to clarify and reply to my first concern, but still I would better explain and discuss this critical point. The authors should better articulate the improvement provided in line 523-527.

Author Response

Answer to reviewer remark 3 

Remark of the reviewer 3

1.I appreciate the efforts of the authors trying to clarify and reply to my first concern, but still I would better explain and discuss this critical point. The authors should better articulate the improvement provided in line 523-527.

Answer to this remark by authors:

We have completely rewritten the improvement provided for the reviewer's first concern (in lines 523-537). This new writing is as follows:

“Finally, we discuss the practical value of our approach. At this point, we could not claim that using our method is always better (in terms of effort or time) than merely trying several configurations directly over the dataset by hand. Nevertheless, we point out two practical reasons for using it: first, it provides a systematic way of having a "default'' configuration, just by computing for the given dataset its statistical signature and then getting a prediction from our already trained system, being these straightforward processes. This default configuration can be used only as an initial reference, to be applied or not, according to the best judgment of the person involved in the process. Second, there could be some very specific scenarios where the application of a method like ours is almost mandatory, like, for instance, in real-time applications where some sensors may fail, or new ones are incorporated. In this situation, our approach would find in real-time the best configuration for the new situation, without the need for any training. Considering that trying out and evaluating every possible configuration does require training, it is clear that a real-time response would not be possible. However, beyond very specific cases like the ones mentioned above, the practical utility of using a general configuration predictor as ours has to be judged by a well-informed decision-maker; a quantitative framework for making such a decision is well outside the scope of this paper”.
